# DAWN: Dual Space Regeneration Attack

## Abstract

The growing use of generative models has intensified the need for watermarking methods that ensure content attribution and provenance. While recent semantic watermarking schemes improve robustness by embedding signals in latent or frequency representations, we show they remain vulnerable even under resource constrained adversarial settings. We present DAWN, a training-free, single-image attack that removes or weakens watermarks without access to the underlying model. By projecting watermarked images onto natural priors across complementary representations, DAWN suppresses watermark signals while preserving visual fidelity. Experiments across diverse watermarking schemes demonstrate that our approach consistently reduces watermark detectability, revealing fundamental weaknesses in current designs. Our code is available at `https://anonymous.4open.science/r/DAWN-567A/`

## 1 Introduction

The rise of generative models such as diffusion models (Rombach et al., 2022), DALL-E (Ramesh et al., 2022) and GANs (Goodfellow et al., 2020) has transformed content creation, enabling realistic images with minimal manual effort. This progress raises concerns about provenance and authenticity, as the line between real and synthetic media blurs. Digital watermarking has emerged as a crucial safeguard, yet adversaries increasingly target watermark forgery and removal to evade provenance tracking. Among watermarking strategies, pixel-domain methods remain the most widely deployed for their simplicity and efficiency. Early algorithms such as STEGASTAMP (Tancik et al., 2020), RIVAGAN (Zhang et al., 2019), and hybrid schemes like DWTDCTSVD (Navas et al., 2008) embed imperceptible signatures in pixels or in frequency-domain coefficients (e.g., wavelet or discrete cosine transform coefficients). These methods withstand common distortions but are fundamentally vulnerable to regeneration attacks, where generative models recreate visually equivalent images without the watermark (Zhao et al., 2024). Such attacks exploit the limited representational depth of pixel- or frequency-space watermarks i.e., how shallowly the watermark is embedded within the multi-dimensional data representation. Because these signals occupy shallow, low-level image statistics (e.g., pixel intensities or a narrow set of DCT/wavelet coefficients) of the representation, adversaries can effectively target and erase them.

To address the weaknesses of pixel- and (low dimensional) frequency-domain methods, watermarking has advanced toward more sophisticated semantic watermarking techniques that embed signals within latent or high-dimensional frequency representations during generation. Approaches such as TREE-RING (Wen et al., 2023), ZODIAC (Zhang et al., 2024), and FREQMARK (Guo et al., 2024) influence global image properties such as composition, texture, and structure, rather than relying on shallow perturbations. By aligning signals with semantic content and leveraging frequency invariances, these approaches improve robustness against post-processing and adversarial manipulations. For instance, TREE-RING shows resilience to regeneration attacks by injecting signals directly into the frequency components of the diffusion model's latent space (Zhao et al., 2024). Nonetheless, new vulnerabilities have emerged. Multi-image steganalysis can exploit consistent high-frequency artifacts across outputs generative models (Müller et al., 2025; Li et al., 2023) to suppress watermarks. These attacks, however, typically assume strong conditions such as access to multiple watermarked samples, knowledge of the watermarking model or generator, or the computational resources for per-image optimization. In contrast, more realistic threat models assume adversaries with only a single watermarked image and no access to the underlying generator, parameters, or watermarking scheme.

Figure 1: Visual pairs of watermarked inputs (top) and outputs on attacking with DAWN (bottom). The watermark is suppressed successfully while the images remain perceptually and semantically consistent.

Building on these limitations, this paper bridges the gap between existing vulnerabilities and *realistic* adversarial scenarios by introducing **DAWN** (**D**ual-domain **A**dversarial **W**atermark **N**ullifier), a projection-based attack framework. Our central insight is that watermarking—whether applied in pixel, frequency, or latent space—inevitably perturbs natural image statistics. By sequentially projecting a single watermarked image back onto natural priors across complementary representations, DAWN selectively suppress watermark signals while preserving visual and semantic fidelity. We realize this principle through three lightweight modules: (i) frequency-domain reconstruction to restore spectral regularities, (ii) semantic refinement to maintain high-level structure, and (iii) perceptual color correction to preserve realism. As illustrated in Figure 1, DAWN effectively removes watermarks while keeping outputs perceptually and semantically consistent with their inputs. Unlike prior attacks, our approach is training-free, single-shot, and model-agnostic.

Our main contributions are:
- A training-free, single-image attack effective across pixel-, frequency-, and latent-space watermarking schemes, without requiring model access or multiple samples (§4).
- Empirical analysis, supported by an intuition (§3, §4), showing why projecting watermarked images onto the natural spectral manifold undermines frequency-based watermarks.
- Comprehensive benchmarking across diverse watermarking methods (§6), exposing fundamental weaknesses and offering guidance for more robust designs.

## 2 RELATED WORK

**Digital Watermarking Techniques:** These approaches differ along two axes: the embedding space (pixel, frequency, latent) and the embedding time i.e., whether the watermark is applied post-generation or injected in-generation during the generative process (in the initial noise or intermediate latent/frequency features). Table 1 summarizes representative methods along these two dimensions. Classical techniques such as DWTDCTSVD (Navas et al., 2008) operate in the frequency domain after image

| Method | Pixel | Frequency | Latent | Time |
|---|---|---|---|---|
| DwtDct (Ingemar et al., 2008) | × | ✓ | × | ○ |
| DwtDctSvd (Navas et al., 2008) | × | ✓ | × | ○ |
| RivaGAN (Zhang et al., 2019) | ✓ | × | × | ○ |
| SSL (Fernandez et al., 2022) | × | × | ✓ | ○ |
| ZoDiac (Zhang et al., 2024) | × | ✓ | ✓ | ● |
| Tree-Ring (Wen et al., 2023) | × | ✓ | ✓ | ● |

Table 1: Categorization of watermarking methods. ● dentoes in-generation, and ○ denotes post-generation.

generation, whereas modern (semantic) schemes like TREE-RING and ZODIAC embed watermarks in latent or frequency features during generation. Consequently, recent methods yield semantic watermarking effects that align with higher-level image structure.

**Attacks on Watermarking Schemes:** The vulnerabilities of watermarking have spurred a diverse set of attack strategies. Regeneration attacks (Zhao et al., 2024) employ diffusion models to reconstruct clean images from watermarked ones, effectively erasing pixel-space signals without model access.

Yet they are far less effective against frequency-domain schemes such as TREE-RING (Wen et al., 2023), since diffusion priors mainly suppress pixel-level perturbations while structured frequency signals often survive. Latent-space manipulation provides another avenue: Müller et al. (2025) demonstrate black-box attacks that iteratively optimize latent representations to erase or redirect watermarks. However, these approaches depend on surrogate models and costly optimization, limiting practicality in real-time or resource-constrained settings. Multi-image steganalysis

In summary, existing approaches either rely on surrogate models, auxiliary datasets, or costly iterative optimization. In contrast, we investigate a more practical adversarial setting: a single-image, training-free, and model-agnostic attack that undermines both pixel- and frequency-domain watermarks by leveraging reconstruction and semantic regeneration.

## 3 WHY PIXEL-ONLY REGENERATION FAILS (AND WHAT WE LEARN)

**Hypothesis:** We believe that pixel-only regeneration methods (e.g., Stable Diffusion `img2img` (Rombach et al., 2022)) are insufficient for suppressing frequency-domain watermarks such as TREE-RING, because they primarily regularize the image in pixel space. In contrast, frequency-domain reconstruction (Xu et al., 2020) will be more effective in weakening watermark signals by directly targeting structured spectral artifacts. We test this hypothesis by comparing diffusion-based regeneration with a frequency-domain UNet reconstruction on TREE-RING watermarked images.

**Approach & Setup:** We start with 20 natural images and embed TREE-RING watermarks to produce samples $\{x_w^{(i)}\}_{i=1}^{20}$. Each image is then reconstructed using two methods: (i) diffusion-based regeneration via a single pass of SD-v2 `img2img` (Rombach et al., 2022), and (ii) frequency-domain reconstruction by applying an $8\times8$ blockwise Discrete Cosine Transform (DCT), denoising the coefficients with a lightweight UNet trained on clean images, and applying the inverse DCT (IDCT). We evaluate results along two axes: (a) *watermark weakening*, measured by TREE-RING detector $p$-values ($p < 0.01$ indicates detection, $p > 0.01$ successful removal), and (b) *perceptual fidelity*, using LPIPS (Zhang et al., 2018) and CLIP similarity (Radford et al., 2021) relative to the original image.

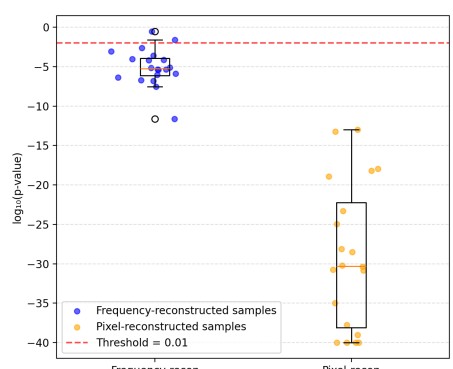

Figure 2: Frequency-domain reconstructions yield higher TREE-RING $p$-values (blue), indicating better watermark retention, whereas pixel-based diffusion (orange) drives values far below the 0.01 threshold.

**Results:** From Figure 2, we observe that the frequency-domain UNet (Xu et al., 2020) produced consistently higher TREE-RING $p$-values than diffusion regeneration (median $5.9 \times 10^{-6}$ vs. $5.1 \times 10^{-31}$, Wilcoxon signed-rank $p = 9.5 \times 10^{-7}$). This indicates stronger weakening of the watermark, though in most cases detection still held ($p < 0.01$). Notably, 10% of frequency-based reconstructions crossed the 0.01 threshold (successful removal), while diffusion reconstructions universally yielded extremely low $p$-values, signifying that the watermark remained strongly detectable and the attack failed. For perceptual fidelity, diffusion regeneration performed better: LPIPS scores were lower (0.10 vs. 0.53) and CLIP similarity higher (0.99 vs. 0.80). Thus, diffusion models regenerate clean, semantically faithful images but fail to remove watermarks, whereas frequency reconstruction can occasionally succeed in removal but at the cost of perceptual quality.

**Takeaways:** These results yield two guiding principles for attack design:

1. *Frequency-Domain Projections are Useful for Suppressing Persistent Watermark Signals.* They are necessary to weaken watermark traces that survive pixel-only regeneration.
2. *Semantic Regeneration is Essential for Preserving Structure and Minimizing Perceptual Artifacts.* It restores high-level content fidelity and reduces visual degradation introduced during watermark removal.

# 4 OUR APPROACH: DAWN

Building on the insights from Section 3, we design DAWN, our Dual-domain Adversarial Watermark Nullifier, a practical single-image attack framework that circumvents the limitations of prior methods relying on surrogate models, multi-sample access, or costly iterative optimization. Guided by two principles, frequency-domain projections to suppress persistent watermark traces and semantic regeneration to preserve perceptual fidelity, DAWN integrates both into a unified pipeline. The pipeline consists of three lightweight stages: (1) frequency-domain reconstruction to restore spectral regularities while weakening watermark signals, (2) semantic refinement to maintain global content structure, and (3) perceptual color correction to align outputs with natural image statistics.

## 4.1 THREAT MODEL

We consider a practical adversarial setting where the attacker has access to only a *single watermarked image* $x_w$. The adversary's goal is to suppress or erase watermark evidence in this image so that it evades detection, while preserving both semantic content and perceptual quality for downstream use (e.g., redistribution without attribution).

The adversary operates under the following assumptions:
- *Knowledge.* The adversary has no access to the watermarking algorithm, embedding key, or detector; no knowledge of the generative model parameters; and no auxiliary watermarked exemplars.
- *Capabilities.* The adversary may manipulate the pixels of $x_w$ using standard tools such as pretrained generative or restoration models, but without costly iterative optimization, surrogate watermarking models, or access to multiple samples. Moderate compute resources are assumed (e.g., a single-pass diffusion or training a lightweight UNet).
- *Scope.* The attack is image-specific rather than universal: it targets only the given watermarked sample, not all future watermarked images.
- *Constraints.* The attack must maintain perceptual fidelity, avoiding visible artifacts that degrade usability of the image.

This *no-box*, *training-free*, and *exemplar-free* regime closely reflects real-world deployment threats, where adversaries typically encounter isolated watermarked images in the wild.

## 4.2 DETAILED APPROACH

**Step 1: Frequency-Domain Reconstruction.** To enable watermark-agnostic suppression of spectral artifacts, we train a lightweight frequency-domain UNet $f_{\text{freq}}$ on a dataset of clean natural images, denoted $\{x_{tr}^{(i)}\}_{i=1}^N$. This dataset is disjoint from test-time watermarked inputs $x_w$ and contains no watermarks; it is only used once, offline, to expose the model to diverse frequency statistics. Each image is first transformed into its $8 \times 8$ blockwise DCT representation:

$$X_{tr}^{(i)} = \text{DCT}(x_{tr}^{(i)})$$

To mimic watermark-like distortions, we generate corrupted inputs $\hat{X}^{(i)}$ by injecting Gaussian noise into structured frequency bands:

$$\hat{X}^{(i)} = X_{tr}^{(i)} + \epsilon_{u,v},$$

where each coefficient is indexed by $(u, v)$ in its $8 \times 8$ block. Noise is selectively added to coefficients satisfying $t_1 \leq u + v < t_2$, which targets a frequency band. This simulates the concentration of watermark energy in perceptually unobtrusive yet robust regions of the spectrum commonly exploited in semantic watermarking. To further enhance robustness, we introduce a *learnable frequency mask* $M$, a sigmoid-gated tensor that adaptively attenuates anomalous frequency regions:

$$\hat{X}_M^{(i)} = M \cdot \hat{X}^{(i)}, \quad \text{with} \quad M = \sigma(\theta),$$

where $\theta$ is a learnable parameter shared across all samples. The UNet is trained with an $\ell_1$ objective to reconstruct the clean frequency map:

$$\mathcal{L}_{\text{rec}} = \frac{1}{N} \sum_{i=1}^N \| f_{\text{freq}}(\hat{X}_M^{(i)}) - X_{tr}^{(i)} \|_1$$

This stage restores natural spectral decay and suppresses watermark-induced anomalies before re-projection into pixel space.

**Step 2: Diffusion-Based Semantic Refinement.** Although frequency reconstruction reduces watermark traces, it can introduce artifacts and degrade fine textures. To recover perceptual realism, we refine the inverse-DCT output with a pretrained `img2img` diffusion model $\mathcal{D}$:

$$\widetilde{x}_{\text{diff}} = \mathcal{D}(\text{IDCT}(f_{\text{freq}}(\text{DCT}(x_w))))$$

This step leverages semantic priors captured by the diffusion model to restore coherent objects, textures, and global structure, while further eroding watermark signals aligned with high-level semantics.

**Step 3: Tone and Color Correction.** Finally, to harmonize global appearance, we apply a lightweight channel-wise mean–variance matching:

$$\widetilde{x}_{\text{final}}^{(c)} = \sigma_c(x_w) \cdot \frac{\widetilde{x}_{\text{diff}}^{(c)} - \mu_c(\widetilde{x}_{\text{diff}})}{\sigma_c(\widetilde{x}_{\text{diff}})} + \mu_c(x_w),$$

where $\mu_c$ and $\sigma_c$ denote the mean and standard deviation of channel $c$. This correction mitigates global color and tone shifts from earlier stages, ensuring that the final output remains both perceptually natural and semantically faithful.

### 4.3 Underneath The Hood

**Goal 1: Maximizing Attack Success.** The first stage can be viewed as a formal projection of the watermarked image onto the natural-image manifold. Formally,

$$\hat{x} = \arg\min_{z \in \mathcal{M}} \|x_w - z\|_2^2,$$

where $\mathcal{M}$ is the natural-image manifold characterized by its $1/f^\alpha$ spectral statistics, with $f$ denoting the spatial frequency magnitude in the 2-D DCT/Fourier plane (e.g., $f = \sqrt{u^2 + v^2}$ for frequency indices $(u, v)$) and $\alpha \approx 1$ for natural images. Because $\mathcal{M}$ inherently penalizes narrow-band spectral spikes, this projection suppresses the annular energy that encodes the watermark. By attenuating these mid-frequency anomalies, Step 1 directly weakens watermark evidence that pixel-only regeneration cannot remove.

**Goal 2: Minimizing Perceptual Gap.** *Step 2* restores global coherence, textures, and object boundaries, ensuring that watermark suppression does not introduce visible artifacts. *Step 3* further harmonizes channel statistics to match natural image distributions, mitigating global shifts from earlier stages. Together, these steps preserve semantic content and visual fidelity. By combining frequency suppression with semantic and perceptual restoration, DAWN simultaneously achieves high attack success against watermarks while maintaining realistic image quality.

## 5 Experimental Setup

**Threat Model and Datasets.** We evaluate under a strict adversarial setting where the attacker is given access to only a single watermarked image $x_w$ per trial. No auxiliary clean images, watermark keys, or model access are available, reflecting a no-box, exemplar-free, and training-free threat regime. For training the frequency-domain UNet, we use the Stable Diffusion Prompts (SDP) dataset, restricted to clean images only. For training the UNet, we use 10k samples from MS-COCO 2017 dataset (Lin et al., 2014) and SDP dataset, consisting of generative outputs paired with prompts, to capture the stylistic variety of real-world diffusion-based synthesis (total of 20k). For evaluation, we randomly select 500 clean images from each dataset and apply the target watermarking schemes to obtain their watermarked counterparts. All reported metrics are computed on these synthetically watermarked evaluation images. Note that the training and evaluation sets are disjoint, ensuring that the frequency-domain UNet never observes watermarked images during training.

**Targeted Watermarking Methods.** We evaluate DAWN across diverse watermarking schemes spanning pixel, frequency, and latent domains, including RivaGAN (Zhang et al., 2019), Dwt-Dct (Ingemar et al., 2008), DwtDctSvd (Navas et al., 2008), SSL Watermarking (Fernandez et al.,

2022), TREE-RING (Wen et al., 2023), and ZODIAC (Zhang et al., 2024). For comparability across methods of different code lengths, we adopt detection criteria from prior work. We use $k = 32$-bit watermarks for DWTDCT, DWTDCTSVD, RIVAGAN, and SSL Watermarking. A watermark is considered detected if at least 23 out of 32 bits are correctly recovered, corresponding to a significance threshold of $p < 0.01$. This standardized setup provides a consistent basis for evaluating DAWN's effectiveness across watermarking schemes.

**Baselines and Comparative Attacks.** To contextualize DAWN's performance, we compare it against representative single-image attacks. Regeneration-based methods (Zhao et al., 2024) use Stable Diffusion img2img (Rombach et al., 2022) to project images back to the pixel manifold without watermark-specific adaptation, while imprint-removal (Müller et al., 2025) iteratively optimizes latent codes with a surrogate generator. Unlike these baselines, DAWN requires neither surrogate models nor costly iterative optimization and still achieves higher success rates.

**Evaluation Metrics.** We evaluate along two primary axes. For *watermark weakening*, we report detector $p$-values (e.g., for TREE-RING), where lower values indicate stronger suppression of watermark evidence (values above $0.01$ correspond to successful detection). For *perceptual and semantic fidelity*, we compute PSNR, SSIM (Wang et al., 2004), and LPIPS (Zhang et al., 2018) between attacked and original images, alongside CLIP similarity (Radford et al., 2021) to capture semantic alignment. To mitigate sensitivity to color shifts, we additionally report luminance-only variants, $SSIM_{lum}$ and $CLIP_{lum}$, which isolate structural and semantic preservation in the luminance channel in YCrCb space.

**Implementation Details.** DAWN is instantiated is an inference only three-stage pipeline. First, we train an $8 \times 8$ blockwise DCT is applied to the input image, and a frequency-domain UNet on mix of MS-COCO and reconstructs spectral regularities from synthetic noisy inputs using an $\ell_1$ loss (c.f. § 4). During training, we sample structured noise with standard deviations in $\{0.1, 0.2, 0.3, 0.4, 0.5, 0.6\}$ and frequency bands $[0, 5]$, $[5, 10]$, and $[10, 15]$ (DCT index ranges) to encourage robustness across low-, mid-, and high-frequency components. Second, the reconstructed image is refined using Stable Diffusion v2 img2img (Rombach et al., 2022), applied in a single pass with frozen weights to restore semantic coherence. Finally, channel-wise mean-variance matching is applied to align tone and color statistics with the original watermarked image. All models are trained offline on clean data, and inference is training-free, single-shot, and independent of the generator or watermarking architecture (i.e., model-agnostic).

## 6 RESULTS

Our evaluation is designed to answer the following questions:
1. How effective is DAWN at suppressing different watermarking schemes?
2. What is the perceptual and semantic fidelity of attacked images?
3. How does DAWN compare against existing attack baselines in terms of success, efficiency, and resource cost?

From our experiments we draw three key insights. First, DAWN reliably suppresses watermarks across both pixel- and semantic-space schemes, achieving $>95\%$ success on classical baselines and 70–90% on semantic methods. Second, it preserves structural and semantic fidelity, with luminance-based metrics ($SSIM_{lum}$, $CLIP_{lum}$) consistently near $0.99$ despite minor hue shifts. Finally, DAWN matches or exceeds optimization-based baselines while requiring only a single forward pass, reducing computational overhead by more than two orders of magnitude.

### 6.1 EFFECTIVENESS ACROSS WATERMARK

We evaluate DAWN on six watermarking schemes introduced in § 5 covering pixel, frequency, and latent domains- using the SDP and MS-COCO evaluation sets. As shown in Table 2, DAWN achieves $>95\%$ success against classical pixel/frequency methods (DWTDCT, DWTDCTSVD, RIVAGAN, SSL) while maintaining high perceptual fidelity (PSNR $\sim$28 dB, CLIP $\sim$0.84). Against semantic schemes, success rates remain high (70% for TREE-RING, 90% for ZODIAC), though with lower PSNR and higher LPIPS, reflecting the tighter coupling of these watermarks with image semantics because its watermark resides in low-frequency latent noise. Notably, luminance-based metrics

(SSIM$_{\text{lum}}$, CLIP$_{\text{lum}}$) remain near 0.99, showing that structural information is preserved even when chromatic deviations occur.

| Metric | TREE-RING | | DWTDCT | | ZODIAC | | DWTDCTSVD | | RIVAGAN | | SSL | |
|---|---|---|---|---|---|---|---|---|---|---|---|---|
| | SDP | MS-COCO | SDP | MS-COCO | SDP | MS-COCO | SDP | MS-COCO | SDP | MS-COCO | SDP | MS-COCO |
| PSNR ↑ | 14.56 | 16.12 | 28.21 | 28.23 | 16.24 | 16.93 | 28.21 | 28.24 | 28.21 | 28.23 | 28.20 | 28.23 |
| LPIPS ↓ | 0.64 | 0.67 | 0.47 | 0.52 | 0.63 | 0.54 | 0.48 | 0.53 | 0.48 | 0.53 | 0.47 | 0.52 |
| SSIM ↑ | 0.46 | 0.45 | 0.54 | 0.50 | 0.45 | 0.56 | 0.54 | 0.50 | 0.55 | 0.51 | 0.54 | 0.50 |
| SSIM$_{\text{lum}}$ ↑ | 0.99 | 0.99 | 0.99 | 0.99 | 0.95 | 0.95 | 0.99 | 0.99 | 0.99 | 0.99 | 0.96 | 0.98 |
| CLIP Similarity ↑ | 0.73 | 0.69 | 0.84 | 0.75 | 0.68 | 0.70 | 0.84 | 0.75 | 0.84 | 0.75 | 0.84 | 0.75 |
| CLIP$_{\text{lum}}$ ↑ | 0.99 | 0.99 | 0.99 | 0.99 | 0.99 | 0.99 | 0.99 | 0.99 | 0.99 | 0.99 | 0.93 | 0.95 |
| Attack Succ. ↑ | 70.2% | 77.4% | 99.8% | 99.8% | 92.2% | 94.4% | 98.40% | 95.80% | 100% | 100% | 95.80% | 97% |

Table 2: Our attack performance across watermarking methods on SDP and MS-COCO datasets.

## 6.2 COMPARISON WITH OTHER ATTACKS

We compare DAWN against regeneration-based and optimization-based baselines. Distortion-based attacks (JPEG, blur, color jitter) are excluded, as prior work shows they fail to reliably remove TREE-RING watermarks (Zhao et al., 2024; Zhang et al., 2024). As shown in Figure 3, DAWN achieves stable success in a single forward pass (takes couple of seconds), while imprint-removal (Müller et al., 2025) requires ∼125 optimization steps to reach comparable performance. We also evaluate a regeneration-based baseline (Zhao et al., 2024) using Stable Diffusion v2, which again fails under our evaluation consistent with the motivating analysis in § 3 showing that pixel-only regeneration cannot suppress frequency-embedded signals such as those in TREE-RING. Qualitatively, Figure 4 shows that while imprint-removal often leaves residual signals or semantic artifacts, DAWN suppresses watermarks more cleanly. Importantly, Y-channel (luminance) images confirm that DAWN preserves structural fidelity, even if hue shifts emerge due to frequency perturbations.

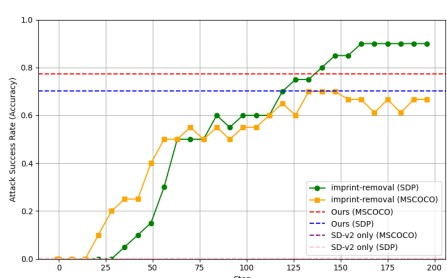

Figure 3: Attack success vs. steps on TREE-RING (SDP, MS-COCO). Our inference-only attack attains high success in one pass; optimization-based imprint-removal(Müller et al., 2025) converges slowly. Regeneration (SD-v2) from Zhao et al. (2024) fails consistently

| | Success↑ | | PSNR↑ | | SSIM↑ | | LPIPS↓ | | CLIP↑ | |
|---|---|---|---|---|---|---|---|---|---|---|
| Variant | MS-COCO | SDP | MS-COCO | SDP | MS-COCO | SDP | MS-COCO | SDP | MS-COCO | SDP |
| – FreqRecon | 27.2 | 27.0 | 7.71 | 7.05 | 0.58 | 0.57 | 0.67 | 0.59 | 0.73 | 0.73 |
| – SemRefine (SD-v2) | 0.0 | 0.0 | - | - | - | - | - | - | - | - |
| – SemRefine (SDXL) | 0.0 | 0.0 | - | - | - | - | - | - | - | - |
| – FreqRecon + colorCorr | 30.0 | 23.8 | 16.99 | 17.2 | 0.67 | 0.69 | 0.57 | 0.51 | 0.77 | 0.76 |
| – FreqRecon + SemRefine (SD-v2) | 61.4 | 65.8 | 7.70 | 7.07 | 0.54 | 0.54 | 0.72 | 0.64 | 0.70 | 0.75 |
| – FreqRecon + SemRefine (SDXL) | 60.2 | 65.2 | 7.83 | 7.03 | 0.56 | 0.55 | 0.72 | 0.65 | 0.71 | 0.69 |
| **DAWN (SD-v2)** | **77.4** | **70.2** | 16.12 | 14.56 | 0.45 | 0.46 | 0.67 | 0.64 | 0.69 | 0.73 |
| **DAWN (SDXL)** | 75.2 | 68.6 | 15.68 | 15.32 | 0.51 | 0.48 | 0.68 | 0.66 | 0.71 | 0.71 |

Table 3: Ablation study of DAWN components on MS-COCO and SDP datasets with TREE-RING watermarks. For modules depending on a generative backbone (SemRefine and its combinations), we report separate rows for SD-v2 and SDXL; backbone-agnostic modules (FreqRecon, FreqRecon + ColorCorr) are reported once.

## 6.3 ABLATION AND COMPONENT ANALYSIS

To understand the role of each module in DAWN, we conduct a detailed ablation on TREE-RING watermarks using both MS-COCO and SDP datasets. Table 3 reports attack success, PSNR, SSIM, LPIPS, and CLIP similarity when selectively retaining frequency-domain reconstruction (FreqRecon), semantic refinement (SemRefine), and color correction (ColorCorr). Removing *frequency-domain reconstruction* leads to the steepest decline in attack

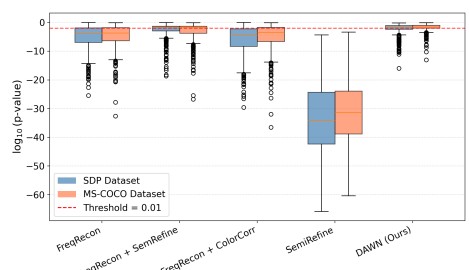

Figure 5: Distribution of detector $p$-values ($\log_{10}$ scale) for DAWN and its ablated variants. Boxes show interquartile range across images from SDP

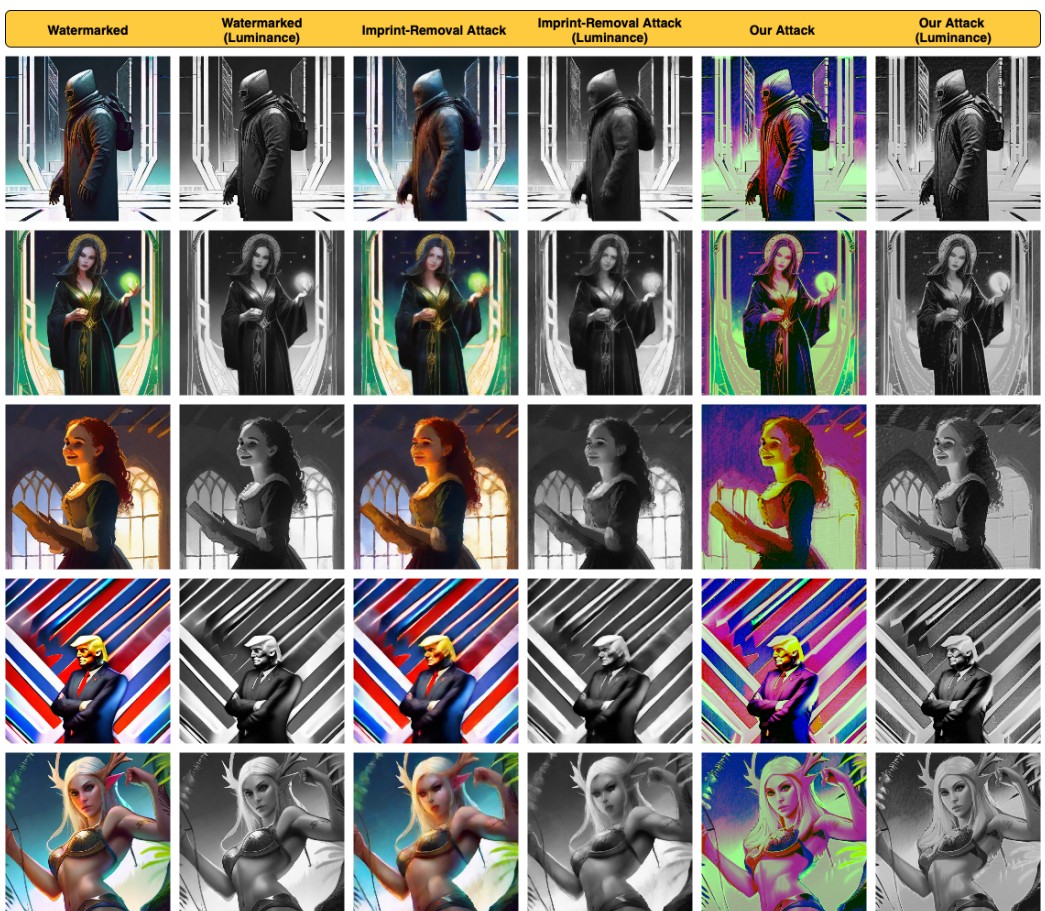

Figure 4: TREE-RING watermarked images (leftmost), results of the Imprint-Removal attack Müller et al. (2025)(third column), and our attack (fifth column), Column second, fourth, and sixth represents Y-channel (luminance) of images in YCbCr space

success-dropping to near zero, clearly confirming that suppressing structured spectral patterns is the primary mechanism for watermark removal. Semantic refinement provides an additional but more modest boost to success by stabilizing high-level structure and semantics, while color correction mainly enhances visual quality by mitigating hue shifts. Overall, the full DAWN configuration achieves the best trade-off between removal strength and perceptual fidelity.

To statistically validate these differences, we compute per-image TREE-RING detector $p$-values for each variant and plot their distributions in Figure 5. Values are shown as $\log_{10}$ of $p$-value; the dashed red line indicates the $0.01$ significance threshold. All DAWN variants remain well below this threshold, but frequency reconstruction removal causes a clear upward shift, and the full DAWN consistently achieves the highest $p$-values. This supports our hypothesis that frequency-domain projection is the dominant mechanism for neutralizing semantic watermarks.

### 6.4 QUALITATIVE OBSERVATIONS

Finally, we visualize few more representative results in Figure 6. DAWN consistently suppresses watermark patterns while preserving both perceptual realism and semantic content. For TREE-RING, which encodes in low-frequency latent noise, we observe residual difficulty in fully erasing the watermark without slight degradation in chromatic fidelity. Nevertheless, these results show that

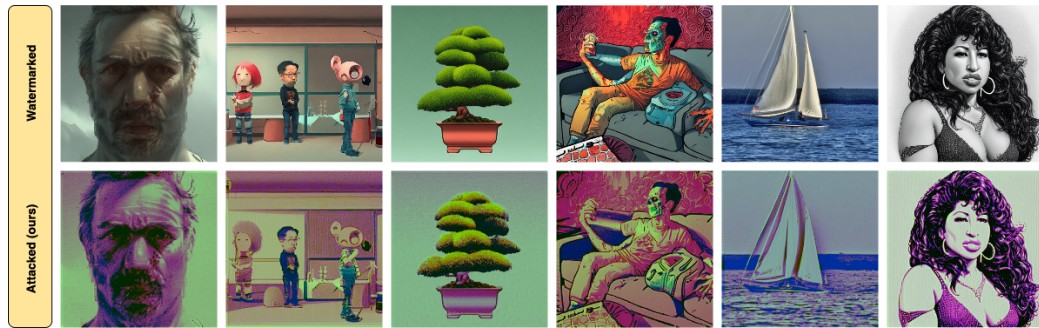

Figure 6: Visual pairs of TREE-RING watermarked inputs (top) and outputs on attacking with DAWNet (bottom). The watermark is suppressed while the images remain perceptually and semantically consistent.

DAWN generalizes across diverse watermarking schemes, highlighting vulnerabilities in both pixel- and frequency-space designs under a realistic single-image, no-box threat model.

## 7 DISCUSSION

DAWN exposes fundamental weaknesses in current generative watermarking by showing that both pixel- and frequency-space watermarks can be removed with a single, training-free forward pass, revealing that many watermark designs share exploitable structural regularities in the frequency domain. These findings suggest that natural-image frequency priors can be leveraged to neutralize watermark signals without access to model parameters, posing a realistic threat to provenance mechanisms. While DAWN preserves structural and semantic fidelity, some outputs exhibit mild chromatic shifts, and our evaluation is confined to Stable-Diffusion family models and common prompt datasets, so generative architectures with substantially different latent structures or future watermarks that interleave multiple frequency bands or semantic layers may reduce its effectiveness. The results highlight the need for watermarking schemes that embed signals across multiple spectral and semantic hierarchies or incorporate cryptographic authentication to remain verifiable under frequency-projection attacks.

## 8 CONCLUSION

We introduced DAWN, a training-free, inference-only attack that suppresses both pixel- and semantic-space watermarks with a single forward pass while preserving structural and semantic fidelity. Comprehensive experiments across six watermarking schemes and two datasets show that DAWN consistently outperforms regeneration and optimization-based baselines, revealing a common frequency-domain vulnerability in current generative watermark designs. These results call for watermarking strategies that jointly exploit spatial, spectral, and semantic cues or integrate cryptographic verification to remain robust against frequency-projection attacks.

## ETHICS STATEMENT

**Use of Large Language Models (LLMs).** Large Language Models were used only for two purposes: (i) to polish writing and improve readability of the manuscript, and (ii) to assist with retrieval and discovery of related work (e.g., helping to locate relevant papers). All conceptual ideas, experimental design, algorithm development, mathematical derivations, and data analysis were conceived, implemented, and verified by the authors. LLMs were not used to generate novel research content, code, or experimental results. Any automated assistance was carefully reviewed and edited by the authors to ensure accuracy and originality, in accordance with the ICLR 2026 guidelines on responsible LLM use.

**Human Oversight and Accountability.** The authors accept full responsibility for every scientific claim and for the correctness of all results presented in the paper. No proprietary or confidential data were provided to external services during the preparation of this work.

**Data and Model Transparency.** This research relies solely on publicly available datasets and open-source generative models: MS-COCO (Lin et al., 2014) and the Stable Diffusion Prompts dataset for clean image training, evaluation, and Stable Diffusion v2/XL as the generative backbone for semantic refinement. The frequency-domain UNet was trained only on clean images.

**Conflict of Interest.** The authors declare no financial or non-financial conflicts of interest related to this work.

## REPRODUCIBILITY STATEMENT

To enable full reproducibility, we provide an anonymized public repository (linked in the abstract) containing: all source code, pretrained frequency-domain UNet checkpoints, configuration files, and scripts to reproduce every experiment. Hyperparameters, random seeds, and detailed evaluation protocols are fully documented, allowing independent groups to replicate the reported results. Our code is available at `https://anonymous.4open.science/r/DAWN-567A/`

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

APPENDIX

In this section we discuss some of the critical analysis.

**R1: On reporting paired PSNR/LPIPS vs. attack-success curves.** Pixel-domain regeneration attacks admit a continuous "strength" parameter (e.g., DDIM noise level, diffusion steps), enabling smooth PSNR/LPIPS–vs–success sweeps. DAWN's frequency-domain module is fundamentally different: the watermark energy appears in different subbands across images (mid, high, or mixed-frequency rings), and our reconstruction some suppresses band frequencies. Because each image has a distinct DCT profile, there is no single scalar parameter that uniformly trades perceptual distortion against suppression strength across the dataset. Artificially introducing a global frequency-scaling sweep would not reflect how watermark energy manifests or how DAWN operates. Thus, the frequency module does not support a comparable parameter sweep.

**R2: Clarification of "single forward pass" and diffusion steps.** DAWN is training-free and contains no iterative optimization loops. A "single forward pass" refers to executing each stage once: (1) frequency-domain reconstruction, (2) one image-to-image diffusion refinement pass, and (3) one deterministic color/tone correction. Step 2 uses a standard $T{=}50$-step DDIM solver for forward corruption and reverse denoising. These are standard inference steps of a pretrained diffusion model and do not constitute iterative re-optimization. Similarly, in Section 3 "a single pass of SD-v2 img2img" refers to a full diffusion purification step, not just the VAE encoder–decoder. We explicitly provide all parameters: we use DDIM with 50 steps, noise strength 0.15, and classifier-free guidance scale 7.5, identical to standard SD-v2 img2img inference settings.

**R3: Tree-Ring watermark application.** Since, TREE-RING is an in-generation watermark our paper does *not* apply TREE-RING post-hoc to clean images. Instead, we follow the official TREE-RING implementation and *generate all watermarked images directly via Stable Diffusion with the* TREE-RING *watermark enabled*. All attacks, including DAWN and baselines, operate exclusively on these already-watermarked images.

**New Experiments on Stronger Watermarks.** To strengthen evaluation, we additionally tested three state-of-the-art watermarking schemes: WAM(Sander et al., 2024) (32 bits), INVISMARK(Xu et al., 2025) (100 bits), and TRUSTMARK (Bui et al., 2023) (100 bits). Results on SDP and MS-COCO are below.

| Metric | WAM | | INVISMARK | | TRUSTMARK | |
|---|---|---|---|---|---|---|
| | SDP | MS-COCO | SDP | MS-COCO | SDP | MS-COCO |
| PSNR ↑ | 28.19 | 28.20 | 28.20 | 28.21 | 28.21 | 28.22 |
| LPIPS ↓ | 0.48 | 0.51 | 0.47 | 0.52 | 0.48 | 0.52 |
| SSIM ↑ | 0.54 | 0.57 | 0.56 | 0.52 | 0.57 | 0.52 |
| $SSIM_{lum}$ ↑ | 0.84 | 0.82 | 0.91 | 0.90 | 0.82 | 0.81 |
| CLIP Sim. ↑ | 0.81 | 0.78 | 0.82 | 0.72 | 0.81 | 0.71 |
| $CLIP_{lum}$ ↑ | 0.88 | 0.86 | 0.89 | 0.85 | 0.88 | 0.84 |
| Attack Succ. ↑ | 85.4% | 81.2% | 89% | 90% | 98.2% | 98.8% |

Table 4: DAWN performance across WAM, INVISMARK, and TRUSTMARK.

**Discussion.** DAWN generalizes effectively to robust watermarking schemes. PSNR and LPIPS remain stable across datasets and watermark types, showing that DAWN does not rely on excessive distortion. CLIP and luminance-CLIP indicate strong semantic and structural preservation. Most importantly, DAWN achieves high removal rates (85–90% on WAM/INVISMARK and $> 98\%$ on TRUSTMARK), demonstrating that frequency-domain reconstruction complements regeneration and remains effective against modern high-capacity watermarks.

**R4: Fairness of the hypothesis test in §3.** Section 3 served as a motivation: frequency perturbations *can* disrupt frequency-based watermarks but offer limited perceptual control because semantic content occurs across different frequency bands. To evaluate fairly, we varied diffusion regeneration strength ($T \in \{2, 3, 4, 5, 10, 50\}$). Even at $T{=}2$, which induces minimal distortion, regeneration achieves higher PSNR, SSIM, and CLIP and lower LPIPS than DAWN. However, these regenerated images

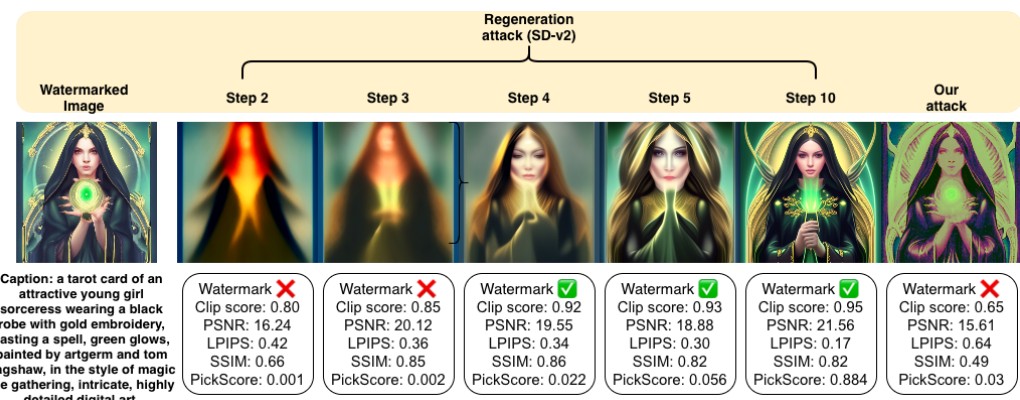

Figure 7: Quality vs. watermark removal for pixel-only regeneration (SD-v2) vs. DAWN. Small-step regeneration ($T{=}2$–5) yields high PSNR/LPIPS/CLIP but the watermark remains. Larger $T$ removes the watermark but causes semantic drift. DAWN achieves effective watermark weakening with better human-preference alignment (PickScore), despite lower pixel-level metrics.

often blur or distort important details and retain the watermark (Fig. 7). Their human-preference alignment PickScore (Kirstain et al., 2023) remains much lower than DAWN's.

DAWN, although scoring lower on pixel-level metrics, consistently achieves higher PickScore and significantly stronger watermark weakening. At comparable perceptual operating points (e.g., $T{=}2$), DAWN removes the watermark while regeneration does not. This shows that PSNR/LPIPS/SSIM and CLIP being biased toward low-frequency similarity, do not reflect semantic fidelity in this setting, while PickScore does.

**Why do PSNR, LPIPS, SSIM, and CLIP all favor the $T{=}2$ regeneration image, even though it is visually less faithful?** All four metrics are biased toward low-frequency preservation and pixelwise similarity rather than semantic or perceptual alignment. At $T{=}2$, diffusion regeneration produces a heavily blurred, low-detail image that remains close to the reference in *pixel-average* sense, which artificially inflates PSNR and SSIM and decreases LPIPS. Blurring reduces high-frequency error by definition, so PSNR and SSIM report a "better" score even though meaningful image content is lost. Similarly, CLIP similarity is dominated by global coarse structure and ignores color distortions and fine spatial detail. Thus, the $T{=}2$ image retains a high CLIP cosine despite exhibiting melted textures, loss of facial features, and poor human-perceived fidelity.

In contrast, DAWN reconstructs and restores high-frequency structure through its frequency-domain module and refinement step. These semantically meaningful details are not rewarded by PSNR/SSIM, which treat high-frequency corrections as "errors," nor fully captured by CLIP. However, they are recognized by preference-aligned metrics such as PickScore, which evaluates human-perceived semantic fidelity, composition, and aesthetic coherence. DAWN therefore scores lower on pixel-level metrics but higher on human-aligned quality criteria while achieving substantially stronger watermark suppression.

