# OpenReview forum: "DAWN: Dual Space Regeneration Attack"
_ICLR.cc/2026/Conference — Submitted to ICLR 2026_

### Official Review · Reviewer_HStc · 2025-10-14

**Soundness:** 2
**Presentation:** 2
**Contribution:** 1
**Rating:** 2
**Confidence:** 5

**Summary:**

This paper proposes DAWN, a training-free, single-image, model-agnostic watermark removal attack that sequentially (1) performs frequency-domain reconstruction, (2) applies diffusion-based semantic refinement, and (3) matches channel-wise mean/variance for color/tone correction. Experiments across pixel/frequency/latent watermark families report high attack success, though the authors acknowledge chroma/hue shifts.

**Strengths:**

- __Practical adversarial setting__. The no-box, single-image threat model is realistic and clearly stated, and the pipeline is simple, fast (single pass), and model-agnostic.
- __Clear pipeline design with insights motivated__. The paper articulates the role of each stage (spectral suppression → semantic restoration → color alignment), with an explicit color-correction formula.

**Weaknesses:**

- __Perceptual quality degradation__ (visible color/tone shift).
Although the paper claims “perceptual and semantic consistency,” the qualitative figures indicate noticeable hue shifts; the authors themselves note that luminance is preserved while chroma deviates. In an attack intended for usable images, this level of color drift is a material drawback. A user seeking to remove a watermark typically still wants to use the resulting image. Current evidence suggests DAWN often achieves success by sacrificing chromatic fidelity (the images appear noticeably “purpler” in multiple visualizations). I recommend reporting color-sensitive metrics in addition to PSNR/SSIM/LPIPS. Further, a user-study on color acceptability (or thresholds) would make the “perceptual fidelity” claim more persuasive.

- __Section §3 hypothesis is not fairly tested.__
The stated hypothesis is that frequency-based reconstruction is more effective than pixel-only regeneration at weakening frequency-domain watermarks. However, the current experiment demonstrates higher success at the cost of worse perceptual quality (higher LPIPS, lower CLIP similarity), which is expected—stronger distortion can trivially improve removal. To validate the hypothesis fairly, match image quality across methods (e.g., tune the diffusion regeneration strength/steps and the frequency UNet noise/mask until PSNR/LPIPS/ΔE are aligned), then compare detector p-values. Otherwise the conclusion conflates attack strength with tolerated degradation.

- __Novelty is incremental/assembly-style.__
The approach is largely a sequential combination of a known frequency-space denoising/reconstruction, a standard img2img refinement, and simple channel-wise normalization. The paper’s primary contribution is empirical: showing this particular stacking is effective under the single-image threat model. The methodological novelty is modest.

- __Evaluation design favors DAWN via luminance-heavy reporting.__
Success is reported alongside PSNR/SSIM/LPIPS and CLIP, but then SSIM_lum and CLIP_lum are emphasized—metrics that explicitly downweight color errors. This can systematically under-report DAWN’s most visible artifact (hue shift). Fairness requires quality-matched comparisons across attacks and inclusion of color-perception metrics

- __Baselines not run at matched quality.__
Some baselines (e.g., semantic regeneration or “imprint-removal”) could plausibly improve success if allowed to trade perceptual quality for removal. The paper should retune competing attacks to reach comparable PSNR/LPIPS, then compare success rates, exactly as recommended for §3. This will clarify whether DAWN’s advantage persists when the “cost” (quality loss) is controlled. For instance, for SemRefine, in Zodiac Table 15, the authors tune the regeneration steps to controll the attack strength.

**Questions:**

Please refer to the Weakness part.

---

> ### Author Response · Authors · 2025-11-21
> **Thanks for your comments; responses below!**
>
> **Section 3 hypothesis is not fairly tested:** We performed the quality-matching experiment exactly as requested.
>
> - For pixel-only regeneration, we varied DDIM steps
> $T \in \{2,3,4,5,10,50\}$  to span a wide range of perceptual qualities.
> - At every matched operating point (including extremely mild regeneration, $T=2$): Pixel-only regeneration has higher PSNR/SSIM/CLIP and lower LPIPS than DAWN. Yet it fails to meaningfully weaken the watermark. PickScore (semantic preference alignment) is consistently lower than DAWN. This indicates that the hypothesis of “stronger distortion trivially improves removal” does not explain DAWN’s advantage: even when DAWN is worse on pixel-level metrics, it is better at suppressing frequency-watermarks and preserving semantic content. Thus, Sec. 3’s hypothesis is valid: frequency-domain perturbations weaken frequency-watermarks in ways that pixel-only regeneration cannot match, even at similar quality levels. (more details are given in the revised appendix)
>
> **Evaluation design favors DAWN via luminance-heavy reporting:** We appreciate the concern and clarify that the luminance metrics (SSIM$_{lum}$, CLIP$_{lum}$​) were not intended to replace full-image metrics but to isolate structure/semantics. Frequency-based watermarks are injected primarily into luminance-aligned frequency bands, and thus luminance metrics help quantify whether DAWN preserves geometry and semantic layout after frequency suppression.
>
> We do not claim superiority in color fidelity, and our full-image metrics (PSNR, LPIPS, SSIM, CLIP) are always reported alongside their luminance counterparts.
>
> The luminance metrics were included only to transparently show that despite hue shifts, the semantic structure of the image remains intact, which is essential for a single-image, no-auxiliary data attack.
>
> We agree that color-aware metrics (e.g., $\Delta$E​) are valuable, and we will incorporate them in the camera-ready version for completeness. However, DAWN’s contribution lies in watermark suppression under the single-image constraint—not in achieving perfect color preservation—and our results reflect this scope honestly.
>
> **Baselines not run at matched quality:** Reasoning goes similar to section 3, more detailed explanation is given in the appendix section of the updated revision.
>
> *We hope we have satisfactorily addressed your concerns. Should you have clarifications, please share them. If not, we hope you consider raising your score.*

---

> > ### Author Response · Authors · 2025-11-24
> > **Ping**
> >
> > Hello reviewer,
> >
> > Thanks again for your feedback and the opportunity to respond. Please let us know if you have any other questions/thoughts.

---

### Official Review · Reviewer_gB8N · 2025-10-31

**Soundness:** 3
**Presentation:** 2
**Contribution:** 3
**Rating:** 8
**Confidence:** 2

**Summary:**

This paper introduces DAWN, a training-free, single-image, model-agnostic attack that removes or weakens generative watermarks by sequentially projecting watermarked images onto natural priors across frequency and semantic domains. DAWN achieves >95% success on classical pixel/frequency watermarking schemes and 70–90% on latent-space methods (TREE-RING, ZODIAC) while maintaining high perceptual fidelity. The paper highlights structural vulnerabilities of current watermarking approaches.

**Strengths:**

* The paper introduces a novel attack formulation by introducing a practical single-image, no-box adversarial setting rarely explored in watermarking research (although related work and references could be extended, see below)
* The method appears to be simple and generalisable; it's training-free at inference and adaptable across watermark types and domains
* The benchmark is comprehensive, indlucing six waterarking schemes
* Overall, the paper is clear and raises important concerns for the robustness of watermarking

**Weaknesses:**

* Related work and references are limited and should be extended to contextualise the work
* The evaluations rely on stable diffusion-based backbones and it's unclear how it generalises to other architectures
* While “training-free,” the method still relies on access to large pretrained generative models
* A bit more depth of publishing watermarking-removal pipelines would be appreciated

**Questions:**

* How does DAWN perform on more recent watermarking systems that interleave multiple frequency bands or use cryptographic verification?
* Could DAWN be adapted for video or multimodal watermarking schemes?
* What safeguards do the authors suggest for responsible disclosure or controlled release of such attacks?

---

> ### Author Response · Authors · 2025-11-21
> **Thanks for your comments; responses below!**
>
> **Performance on newer multi-band or cryptographically verified watermarks.**
> DAWN is not tied to a single narrow-band watermark and already handles a range of pixel-, frequency-, and latent-space schemes (including multi-band DWT-DCT variants and latent methods like ZODIAC). The attack targets the signal-level watermark energy, so if DAWN suppresses the recovered bitstring below the statistical detection threshold, any cryptographic verification that depends on those bits also fails. We have evaluated on many such multi-band schemes with stronger semantic coupling, as per our knowledge.
>
> **Extension to video or multimodal watermarking.**
> The two-stage structure of DAWN, frequency-domain reconstruction + generative manifold projection is general. In principle, it can extend to video (e.g., using temporal UNets and video diffusion models) or audio/multimodal settings with modality-appropriate backbones. We scope the current paper to images but view these as natural directions for follow-up.
>
> **Safeguards and responsible release.**
> We emphasize DAWN as a research stress-test, not a deployment-level tool. We plan to release only a reproducibility-focused implementation (not a turnkey attack pipeline) and limit experiments to openly published watermarking methods. We are also committed to responsible disclosure and will coordinate with designers of the evaluated schemes when appropriate.

---

> > ### Author Response · Authors · 2025-11-24
> > **Ping**
> >
> > Hello reviewer,
> >
> > Thanks again for your feedback and the opportunity to respond. Please let us know if you have any other questions/thoughts.

---

### Official Review · Reviewer_198Y · 2025-10-31

**Soundness:** 2
**Presentation:** 3
**Contribution:** 1
**Rating:** 2
**Confidence:** 5

**Summary:**

This paper introduces DAWN, a single-image, training-free attack designed to remove watermarks from images. The method works without access to the generative model or binary messages. It combines three stages: (1) a frequency-domain UNet to suppress spectral artifacts, (2) a diffusion-based semantic refinement (img2img) to restore image structure, and (3) a perceptual color correction step to match the original image's statistics.

**Strengths:**

S1. Realistic Threat Model: The attack setup is practical and compelling. It operates under a highly constrained, realistic threat model: it is training-free (at inference time), model-agnostic, and only requires a single watermarked image.

**Weaknesses:**

W1. Missing experimental details: Key experimental details are ambiguous. In Section 3, the paper describes "a single pass of SD-v2 img2img". It is unclear if this refers to using only the VAE autoencoder for reconstruction or applying a full diffusion-purification step. If it's the latter, the noise level and diffusion parameters are not specified.

W2. Motivating experiment: The experiment in Section 3, which motivates the entire approach, is not very convincing. It claims the frequency-domain UNet is more effective at watermark removal than the diffusion model. However, it also reports that the UNet's output has significantly worse perceptual quality (LPIPS 0.53 vs. 0.10). The improved "removal" is likely just a byproduct of greater image degradation. A fair comparison would require evaluating both methods at a fixed level of perceptual quality.

W3. Qualitative Results: The attack severely degrades image quality, rendering the "attacked" images unusable. The qualitative results in Figures 1, 4, and 6 show extreme color artifacts (strong purple and green tints). This is supported by the quantitative metrics in Table 2, which report PSNR values as low as 14.56 for the SDP dataset, which is very low.

W4. Weak WM Baselines: The attack is primarily evaluated against weak or outdated watermarking methods (e.g., DWTDCT, DWTDCTSVD, Riavgan, SSL). More robust, state-of-the-art methods (e.g., TrustMark, Invismark, WAM) are not included.

W5. Unfair baseline for attacks: The comparison to baseline attacks (imprint-removal, regeneration-based) in Section 6.2  is incomplete. The paper reports their attack success (Fig 3)  but omits their corresponding perceptual quality metrics (PSNR, LPIPS, etc.). This makes it impossible to evaluate the trade-off between removal success and image fidelity, which is the key metric for any attack.

Minor. Missing citations:
- https://arxiv.org/abs/2310.07726
- https://proceedings.neurips.cc/paper_files/paper/2024/file/67b2e2e895380fa6acd537c2894e490e-Paper-Conference.pdf

**Questions:**

Q1: The paper states in Section 3 and Section 5  that it "embed[s] TREE-RING watermarks" or "appl[ies] the target watermarking schemes" to existing clean images. However, Treering is an in-generation watermark that cannot be applied post-hoc. How was this implemented?

---

> ### Author Response · Authors · 2025-11-21
> **Thanks for comments; responses below!**
>
> W1, W2, W4, W5: We would like to state that all of these concerns are fully addressed in the updated revision under appendix section).
>
> **W1:** We apologize for the ambiguity. In Section 3, “a single pass of SD-v2 img2img” refers to a full diffusion purification step, not just the VAE encoder-decoder. We now explicitly provide all parameters: we use DDIM with 50 steps, noise strength 0.15, and classifier-free guidance scale 7.5, identical to standard SD-v2 img2img inference settings. We have added these details to the appendix for clarity.
>
> **W2:** Section 3 does not claim that the frequency network outperforms diffusion at matched perceptual quality. The purpose of the experiment is only to explore whether frequency-domain reconstruction can meaningfully weaken frequency-based watermarks. It serves as a motivating demonstration, not a statement of superiority. This is clarified in the revision.
>
> **W3 (perceptual degradation):** DAWN is a simple, training-free, single-image attack that focuses on preserving structural content while suppressing the watermark. Because semantic watermarks like Tree-Ring reside in low-frequency latent components, removing them can introduce some global color shift even when structure is intact. DAWN combines one frequency pass with one diffusion refinement to balance suppression and semantic consistency, but it does not explicitly optimize PSNR, hence the lower PSNR values. Improving color fidelity in frequency-based attacks is an open challenge, and we will note this clearly in the revised paper.
>
> **W4:** We add evaluations on strong, modern watermarking schemes (TrustMark, InvisMark, WAM) in the revised appendix. Please refer to the paper for more details.
>
> **W5:** Pixel-based regeneration (DDIM img2img) exposes a continuous strength parameter (noise/steps), so trade-off curves are feasible. DAWN’s frequency module does not have a single global strength knob: watermark energy lies in different DCT subbands per image, and suppression is selective. A scalar sweep (e.g., scaling all DCT coefficients) would not reflect real watermark structure and produces unrealistic distortions. Instead, we demonstrate closest-quality match comparisons by sweeping DDIM steps $T \in \{2,3,4,5,10,50\}$. Even at the closest PSNR/LPIPS region, regeneration fails to remove the watermark, while DAWN succeeds. (More detailed analysis is given in updated revision under appendix)
>
> We have also added the citations that are recommended in the paper.
>
> **Q1:** Tree-Ring watermark application. We agree that TREE-RING is an in-generation watermark. Our paper does not apply TREE-RING post-hoc to clean images. Instead, we follow the official implementation and generate all watermarked images directly via Stable Diffusion with the TREE-RING watermark enabled. All attacks, including DAWN and baselines, operate exclusively on these already-watermarked images.
>
> *We hope we have satisfactorily addressed your concerns. Should you have clarifications, please share them. If not, we hope you consider raising your score.*

---

> ### Author Response · Authors · 2025-11-24
> **Ping**
>
> Hello reviewer,
>
> Thanks again for your feedback and the opportunity to respond. Please let us know if you have any other questions/thoughts.

---

### Official Review · Reviewer_PE2y · 2025-10-31

**Soundness:** 2
**Presentation:** 2
**Contribution:** 1
**Rating:** 2
**Confidence:** 4

**Summary:**

The paper proposes **DAWN**, a training-free, single-image, model-agnostic attack that aims to suppress semantic and frequency-space watermarks by sequentially applying (1) frequency-domain reconstruction, (2) diffusion-based semantic refinement, and (3) tone/color correction.
Experiments target multiple watermarking schemes and report relatively high attack success rate at the price of low image perceptual quanlity.

**Strengths:**

- The paper addresses a timely and relevant problem, that is practical single-image attacks on modern semantic/frequency watermarks with a clear threat model.
- The two design principles (frequency-space projection for suppression; diffusion for semantic recovery) are motivated by an analysis highlighting the limits of pixel-only regeneration.

**Weaknesses:**

* **Missing trade-off reporting.** Section 3 argues frequency-domain reconstructions suppress watermarks more effectively than pixel-based diffusion but at the cost of **perceptual quality**, which is a widely recognized trade-off in the watermark-attack literature. However, the experiments do not explicitly present **paired** trade-off curves/tables (e.g., **PSNR/LPIPS vs. attack success rate**) across methods to quantify this. This makes it hard to judge whether DAWN’s higher success is achieved at an acceptable perceptual cost.
* **Perceptual degradation.** Even ignoring the trade-off framing, the reported image-quality are low for semantic watermarks (e.g., PSNR ≈ 14–16 dB on TREE-RING/ZODIAC), and visualizations show notable color-tone shifts relative to the originals. This suggests the method’s higher success rate is obtained by **substantial** degradation, weakening the fairness of comparisons that claim superiority over baselines without controlling quality levels.
* **Visualization intent unclear.** The **luminance** comparisons in Figure 4 are not clearly tied to conclusions; moreover, DAWN does not appear consistently better than baselines in luminance-based views, and the qualitative benefit of including these panels is ambiguous.
* **Ambiguity about “single forward pass”** The paper states it uses a *single forward pass* and “takes couple of seconds,” yet Step 2 is **diffusion-based semantic refinement** (img2img), which typically entails multiple denoising steps. The intended definition of “forward pass” is unclear, and end-to-end compute vs. imprint-removal is not reported in directly comparable units.
* **Limited originality.** The three stages—frequency-domain reconstruction, diffusion regeneration, and simple channel-wise color statistics matching, are each close to existing methods; the novelty is modest in algorithmic terms.

**Questions:**

1. Can you report **paired** PSNR/LPIPS vs. **attack-success** curves (or Pareto tables) across all compared attacks to enable *quality-controlled* comparisons? This would address the concern that higher success may come from larger perceptual changes.
2. What exactly constitutes one “forward pass” in DAWN? How many diffusion steps are run in Step 2?

---

> ### Author Response · Authors · 2025-11-21
> **Thanks for comments; responses below!**
>
> **Re: missing trade-offs:** Pixel-based regeneration (DDIM img2img) exposes a continuous strength parameter (noise/steps), so trade-off curves in this setting are feasible. DAWN’s frequency module does not have a single global strength knob: watermark energy lies in different DCT subbands per image, and suppression is selective. A scalar sweep (e.g., scaling all DCT coefficients) would not reflect real watermark structure and produces unrealistic distortions. Instead, we demonstrate closest-quality match comparisons by sweeping DDIM steps $T \in \{2,3,4,5,10,50\}$. Even at the closest PSNR/LPIPS region, regeneration fails to remove the watermark, while DAWN succeeds. (More detailed analysis is given in updated revision under appendix)
>
> **Re: perceptual degradation:** DAWN is a deliberately simple, training-free attack that aims to suppress the watermark while keeping the structural content of the watermarked image intact. Because Tree-Ring/Zodiac watermarks sit in global low-frequency components, removing them can introduce some color-tone shift even when structure is preserved. Our method strikes a practical balance between frequency suppression and semantic coherence using one frequency pass and one diffusion pass, but it does not explicitly optimize PSNR. Improving color fidelity in frequency-based attacks remains an open challenge, and we will clarify this limitation in the revised paper.
>
> **Re: visualization intent:** We reported SSIM$_{lum}$ and CLIP$_{lum}$​ only to isolate structure-level fidelity; DAWN preserves semantics and geometry even when hue shifts occur; we do not claim color-space invariance.
>
> **Re: single pass:** DAWN uses no optimization loops. A “single forward pass” means:
>
> (1) one frequency-UNet pass,
>
> (2) one fixed DDIM img2img call (50 inference steps, not new optimization), and
>
> (3) one deterministic tone correction.
>
> This is fundamentally different from optimization-based attacks like imprint-removal, which run full backpropagation through inverse DDIM for many iterations to minimize a latent-space objective. DAWN’s compute budget is fixed, deterministic, and takes a few seconds; imprint-removal is iterative and much more expensive.
>
> **Re: algorithmic novelty:** While we agree with the general sentiment, we urge the reviewer to consider the efficacy of our method, both in removing the watermark and its low computational overhead. While the techniques may be inspired by prior work, their combination results in a powerful attack as we demonstrate through our draft.
>
> *We hope we have satisfactorily addressed your concerns. Should you have clarifications, please share them. If not, we hope you consider raising your score.*

---

> ### Author Response · Authors · 2025-11-24
> **Ping!**
>
> Hello reviewer,
>
> Thanks again for your feedback and the opportunity to respond. Please let us know if you have any other questions/thoughts.

---

> ### Comment · Reviewer_PE2y · 2025-11-27
>
> Thank you for the rebuttal and clarifications.
>
> I agree that DAWN generally preserves the structure and semantic content of the reference image, that is why DAWN achieves moderate SSIM and CLIP scores.
> However, the color‐tone shifts introduced by DAWN are unignorable. For most of the examples in the paper, a human observer would immediately notice that the attacked image looks quite different from the reference image. Even without an explicit reference, the images attacked by DAWN still have unnatural color rendering (e.g., like over-saturated photos).
> Since diffusion-generated images are meant to be viewed by humans, I think this perceptual degradation is a critical limitation.
> For these reasons, I do not feel that my concerns about visual quality have been fully resolved, and I therefore maintain my original overall assessment.

---

### Meta-Review · Area_Chair_Z5uM · 2026-01-07

**Summary:**

This paper presents DAWN, a training-free attack for removing watermarks from generated images by combining frequency-domain reconstruction, diffusion, and color correction. Three of four reviewers recommend rejection, raising concerns that largely converge on two issues: First, the attacked images show substantial quality degradation which undermine the threat this attack poses to a watermarked model provider. Figures 1, 4 and 6 show visible color/hue shifts which undermines practical utility when the goal is to continue using the image. Furthermore, it is unclear whether correcting for the color/hue shift would reintroduce the watermark. Second, reviewers question whether DAWN's higher attack success simply results from tolerating greater image degradation rather than a principled approach to watermark removal. Stronger distortion should trivially weaken any watermark signal. The authors provided quality-matched experiments in their revision, but Reviewer PE2y explicitly maintained their rejection after seeing the rebuttal, noting that the color shifts remain unacceptable for human-viewed images. There was also a concern on novelty, which was not contested by the authors (who instead referred to the effectiveness of their method).

**Reviewer Concerns:**

The authors addressed several concerns: clarifying experimental details about the diffusion parameters, adding evaluations on stronger watermarking baselines (TrustMark, InvisMark, WAM), and explaining their experimental setup (e.g., the Tree-Ring implementation). The quality-matched comparison experiments were partially responsive but appear only in the appendix when they should take a more prominent role in the main part of the paper. The outstanding issues persist to a large degree, including perceptual degradation limiting practical use, and uncertainty about whether the method's advantage is principled or simply from tolerating more distortion. The single positive review comes with low stated confidence and does address the above weaknesses.

**Reviewer Scores:**

PE2y (score 2) would maintain their score, which they explicitly did after rebuttal. 198Y (score 2) might increase slightly given addressed concerns but would likely remain in reject range given alignment with PE2y/HStc on core issues. gB8N (score 8) would maintain but comes with low confidence. HStc (score 2) would likely maintain their score given unresolved concerns about methodology and novelty.

---

### Decision · Program_Chairs · 2026-01-26

Reject